# Comparisons of Post-Load Glucose at Different Time Points for Identifying High Risks of MASLD Progression

**DOI:** 10.3390/nu17010152

**Published:** 2024-12-31

**Authors:** Long Teng, Ling Luo, Yanhong Sun, Wei Wang, Zhi Dong, Xiaopei Cao, Junzhao Ye, Bihui Zhong

**Affiliations:** 1Department of Gastroenterology, The First Affiliated Hospital, Sun Yat-sen University, No. 58 Zhongshan II Road, Yuexiu District, Guangzhou 510080, China; tenglong@mail2.sysu.edu.cn (L.T.); luol27@mail2.sysu.edu.cn (L.L.); 2Department of Clinical Laboratory, The First Affiliated Hospital, Sun Yat-sen University, No. 183 Huangpu East Road, Huangpu District, Guangzhou 510080, China; sunyh@mail.sysu.edu.cn; 3Department of Ultrasound, The First Affiliated Hospital, Sun Yat-sen University, No. 58 Zhongshan II Road, Yuexiu District, Guangzhou 510080, China; wangw73@mail.sysu.edu.cn; 4Department of Radiology, The First Affiliated Hospital, Sun Yat-sen University, No. 58 Zhongshan II Road, Yuexiu District, Guangzhou 510080, China; dongzh7@mail.sysu.edu.cn; 5Department of Endocrinology, The First Affiliated Hospital, Sun Yat-sen University, No. 58 Zhongshan II Road, Yuexiu District, Guangzhou 510080, China; caoxp@mail.sysu.edu.cn

**Keywords:** metabolic dysfunction-associated steatotic liver disease, oral glucose tolerance test, post-load glucose, clinical outcomes, liver fibrosis, liver steatosis, atherosclerotic cardiovascular disease, alanine aminotransferase

## Abstract

**Background**: The 1-h post-load plasma glucose was proposed to replace the current OGTT criteria for diagnosing prediabetes/diabetes. However, it remains unclear whether it is superior in identifying progressive metabolic dysfunction-associated steatotic liver disease (MASLD), and thus we aimed to clarify this issue. **Methods**: Consecutive Asian participants (non-MASLD, *n* = 1049; MASLD, *n* = 1165) were retrospectively enrolled between June 2012 and June 2024. CT was used to quantify liver steatosis, while the serum liver fibrotic marker was used to evaluate liver fibrosis. **Results**: Compared with those with normal levels of both 1-h post-glucose (1hPG) and 2-h post-glucose (2hPG), patients with MASLD showed a significant positive association between elevated 1hPG levels and moderate to severe liver steatosis (odds ratio [OR] = 2.19, 95% confidence interval [CI]: 1.13–4.25, *p* = 0.02]. Elevated levels of both 1hPG and 2hPG were associated with an increased risk of liver injury (OR = 2.03, 95% CI: 1.44–2.86, *p* < 0.001). Elevated 2hPG levels with or without elevated 1hPG levels were associated with liver fibrosis (OR = 1.99, 95% CI: 1.15–3.45, *p* < 0.001; OR = 2.72, 95% CI: 1.79–4.11, *p* < 0.001, respectively). Additionally, either 1hPG or 2hPG levels were associated with atherosclerosis, revealing significant dose-dependent associations between glucose status and atherosclerosis risk (OR = 2.77, 95% CI: 1.55–4.96, *p* < 0.001 for elevated 1hPG; OR = 2.98, 95% CI = 1.54–5.78, *p* = 0.001 for elevated 2hPG; OR = 2.41, 95% CI = 1.38–4.21, *p* = 0.001 for elevated levels of both 1hPG and 2hPG). The areas under the ROC for predicting steatosis, liver injury, liver fibrosis, and atherosclerosis were 0.64, 0.58, 0.58, and 0.64 for elevated 1hPG (all *p* < 0.05) and 0.50, 0.60, 0.56, and 0.62 for elevated 2hPG (all *p* < 0.05), respectively. **Conclusions**: These findings underscore the necessity for clinicians to acknowledge that the screening and management of MALSD requires the monitoring of 1hPG levels.

## 1. Introduction

Metabolic dysfunction-associated fatty liver disease (MASLD) is one of the most prevalent chronic liver diseases, affecting nearly one-third of the population [1]. MASLD was renamed from non-alcoholic fatty liver disease (NAFLD) in 2023, and several studies have reported that approximately 99% of patients with NAFLD meet the criteria for MASLD [2,3]. MASLD progression involves subsequent hepatic fibrosis and extrahepatic metabolic complications of obesity, insulin resistance, and type 2 diabetes mellitus (T2DM) [4], burdening the global health system. Epidemiologic evidence indicates that approximately 20~30% of individuals with MASLD progress to metabolic dysfunction-associated steatohepatitis, which is characterized by more severe steatosis, inflammation, and hepatocyte ballooning renal failure [5], and about 20% of them advance to end-stage liver disease, potentially necessitating liver transplantation. Notably, atherosclerotic cardiovascular disease and extrahepatic cancers are the leading causes of mortality among MASLD patients. There is an urgent need for the early identification of people at risk of developing progressive MASLD, allowing for early intervention to handle its comorbidities and complications [6,7].

The current MASLD diagnostic criteria of metabolic dysfunction include fasting blood glucose (FBG), a random blood glucose level over 11.1 mmol/L, and glycated hemoglobin (HbA1c) and plasma glucose measured 2 h after obtaining a 75 g glucose load using the oral glucose tolerance test (OGTT). However, individuals with irregular blood glucose levels may remain undiagnosed for T2DM due to normal fasting glucose levels, resulting in a concealed at-risk group that could be overlooked [8]. Emerging studies and the International Diabetes Federation support establishing 1-h post-glucose (1hPG) ≥155 mg/dL (8.6 mmol/L) as a reliable marker for predicting future diabetes and metabolic complications, such as MASLD. A recent study with sample size of 107 patients from Italy showed a positive correlation between steatosis severity determined using a controlled attenuation parameter, with 1hPG but not 2-h post-glucose (2hPG) [9]. A cross-sectional CATAMERI study (*n* = 2335, conducted in Italy and USA) demonstrated that normal FBG levels with high 1hPG was associated with higher risks of concurrent intermediate/advanced liver fibrosis (odds ratio [OR] = 1.44, 95% confidence interval [CI]: 1.01–2.06), whereas FBG, 1hPG, 2hPG, and HbA1c presented similar diagnostic value in predicting liver fibrosis risks [10]. However, all these findings were derived from Gaussian processes, and significant differences in prognoses exist for MASLD among different races and ethnicities [11]. Therefore, assessing the glucose status 0, 1, and 2 h after conducting an OGTT and determining its association with liver steatosis, inflammation, liver fibrosis, and even atherosclerotic cardiovascular disease in Asians may enhance early screening strategies and targeted interventions for MASLD.

Therefore, we conducted a retrospective hospital-based cohort study of MASLD patients to compare the serum levels of FBG, 1hPG, and 2hPG in predicting the risk of liver severity and atherosclerosis in MASLD patients.

## 2. Materials and Methods

### 2.1. Study Design and Population

A cross-sectional design was utilized in this study and it was conducted at the First Affiliated Hospital of Sun Yat-sen University, recognized as the largest institution for fatty liver treatment in southern China. This study adhered to the principles outlined in the Declaration of Helsinki, and its approval was obtained from the hospital’s ethics committee (approval number: [2014] 112). Written informed consent was obtained from all subjects.

We prospectively and consecutively recruited study participants from June 2012 to June 2024, including individuals aged 18 to 75 years who were suspected of having fatty liver disease. Factors for inclusion included a family history of diabetes and fatty liver disease, self-reports of right upper abdominal discomfort, significant weight gain, and unexplained fatigue. The study protocol was registered in 2014 and defined as NAFLD. The collected variables included all necessary components for diagnosing MASLD. In the analysis, participants were diagnosed according to the EASL 2024 criteria for MASLD, which require the identification of liver steatosis through abdominal ultrasonography imaging and the fulfillment of at least one of the following cardiometabolic criteria: (1) a body mass index (BMI) of 23 kg/m^2^ or greater, or a waist circumference (WC) exceeding 90 cm for men and 80 cm for women; (2) a FBG level of 5.6 mmol/L or higher, a diagnosis of T2DM, or the use of medication for T2DM; (3) blood pressure readings of 130/85 mmHg or higher, or the administration of antihypertensive medications; (4) plasma triglyceride (TG) levels of 1.70 mmol/L or above, or the current use of lipid-lowering treatments; and (5) plasma high-density lipoprotein cholesterol (HDL-C) levels of ≤1.0 mmol/L for males and ≤1.3 mmol/L for females, or the use of lipid-lowering agents [12].

All study participants underwent comprehensive anthropometric assessments, laboratory tests, and imaging examinations, including a complete OGTT at 0 h, 1 h, and 2 h. The study excluded participants with serious medical conditions, including cancers, heart attacks, gastrointestinal bleeding, and liver or renal failures. Additionally, individuals who were pregnant or nursing, as well as individuals with a history of significant alcohol consumption (exceeding 210 g per week for men and 140 g or more for women), were not included. Furthermore, anyone diagnosed with diabetes and currently taking antidiabetic medications prior to the study’s commencement was also excluded. Any medical condition (such as pheochromocytoma, Cushing’s disease, or acromegaly) or treatment (including fibrates, metformin, thiazolidinediones, or statins) that could potentially affect glucose or lipid metabolism was deemed an exclusion criterion. Moreover, individuals with splenomegaly, as identified by abdominal ultrasound or CT imaging, were also excluded.

### 2.2. Clinical and Laboratory Assessments

Demographic details, past illnesses, medication use, and information regarding nicotine and alcohol consumption were collected via a structured questionnaire as in a previous report [13]. Each participant underwent a comprehensive physical examination to assess body weight, height, WC, the waist-to-hip ratio (WHR), and blood pressure. A BMI of 23 kg/m^2^ or greater was established as the criterion for classifying individuals as overweight [12]. Participants with blood pressure readings of 130/85 mmHg or higher were classified as having hypertension [14]. For the diagnosis of T2DM, the diagnostic thresholds included an FBG level of 7.0 mmol/L or greater, or a 2hPG level of 11.1 mmol/L or greater [15].

All participants underwent a 75 g OGTT following an overnight fast, wherein blood samples were collected immediately before glucose administration and at 60 min postprandially, for up to 120 min. Biochemical parameters were assessed using an Abbott c8000 Automatic Biochemistry Analyzer (Abbott, Green Oaks, IL, USA). These parameters included fasting serum insulin (FINS), HbA1c, FBG, 1hPG, 2hPG, total cholesterol (CHOL), TG, HDL-C, low-density lipoprotein cholesterol (LDL-C), uric acid (UA), albumin, and liver enzymes, including alanine aminotransferase (ALT), aspartate aminotransferase (AST), and alkaline phosphatase. The homeostasis model assessment of insulin resistance (HOMA-IR) was calculated using the formula [FINS (μU/mL) × FBG (mmol/L)]/22.5, with a threshold of 2.5 for determining insulin resistance [16]. The homeostasis model assessment of β-cell function (HOMA-β) is defined as 20 × fasting insulin/(fasting glucose—3.5), with a cutoff value of 100% for assessing insulin sensitivity [17]. The cut-off values for FBG, 1hPG, and 2hPG were set at 5.6 mmol/L, 8.6 mmol/L, and 7.8 mmol/L [12,18], respectively. Study participants were categorized into four groups based on their glucose levels at 1 h and 2 h following the OGTT. The first group was characterized by 1hPG and 2hPG levels within normal ranges. The second group had elevated 1hPG levels and normal 2hPG levels. The third group exhibited elevated 2hPG levels with normal 1hPG levels. Finally, the fourth group represented cases where both glucose measures were elevated.

### 2.3. Liver Steatosis Assessments

Abdominal ultrasonography was employed to identify fatty liver in each patient by recognizing specific radiological indicators, including variations in echo patterns between the liver and kidney, potential attenuation of the ultrasound beam in the posterior region, obscuration of vascular structures, delineation of the gallbladder wall, and difficulties in clearly visualizing the diaphragm [19]. In addition, non-enhanced abdominal computed tomography (CT) was utilized to measure liver fat content in a subset of MASLD patients [20]. To obtain hepatic Hounsfield unit (HU) values, three circular regions of interest, each measuring 100 mm^2^, were carefully selected in the axial view of the imaging results. The CT attenuation values were subsequently utilized to classify the severity of steatosis, with predefined thresholds for mild, moderate, and severe steatosis set at 57, 40, and 23 HU, respectively. In this study, moderate and severe liver steatosis was defined by CT attenuation values below 40 HU [21].

### 2.4. Assessment of Hepatic and Extrahepatic Complications

The FIB-4 index is calculated using the following formula: age × AST (IU/L)/[platelet count (×10^9^/L) × √ALT (IU/L)]. A FIB-4 index over 1.3 indicates the presence of significant liver fibrosis [22]. Clinical events related to atherosclerosis were identified using CT angiography, which employed the coronary artery calcium (CAC) score. Furthermore, CAC severity was categorized into four groups according to scores of 0, 0–100, 100–399, and >399 representing as stage normal, mild, moderate, and severe CAD risk. Additionally, the threshold for inclusion in the atherosclerosis group was set at a CAC score of 100 or above [23]. Liver injury was defined as ALT levels exceeding 33 U/L in males and 25 U/L in females [12].

### 2.5. Statistical Analysis

Continuous variables with a normal distribution are reported as the mean and standard deviation. In contrast, continuous variables that do not follow a normal distribution are expressed as the median and interquartile range. One-way ANOVA and the Kruskal–Wallis test were employed to conduct analyses involving multiple comparisons. For categorical variables, the chi-square test was utilized to analyze differences among the groups. Logistic regression was used to assess the relationship between blood glucose status and the risk of clinical outcomes, including liver steatosis, liver injury, liver fibrosis, and atherosclerosis. Statistical analyses were performed using SPSS Statistical Software (version 26.0, IBM Corp, Armonk, NY, USA). Two-sided *p* values less than 0.05 were considered significant.

## 3. Results

### 3.1. Baseline Characteristics

A total of 2214 participants, comprising 1165 MASLD patients and 1049 non-MASLD patients, were included in this study; 584 participants underwent unenhanced abdominal CT to evaluate the degree of liver steatosis. The clinical characteristics of the study subjects are detailed in Table 1. The two groups were comparable in terms of age and platelet levels; however, the MASLD group presented a higher proportion of males (52.2% vs. 41.9%, *p* < 0.001), and a worsening metabolic profile, characterized by higher levels of BMI, WHR, FBG, 1hPG, 2hPG, FINS, HOMA-IR, CHOL, TG, LDL-C, and UA, and a higher proportion of T2DM alongside lower HDL-C levels (all *p* < 0.001). Additionally, MASLD patients exhibited significantly elevated levels of liver enzymes and a higher FIB-4 index compared to the non-MASLD population (all *p* < 0.05).

Among the MASLD patients, serum levels of FBG, 1hPG, 2hPG, HDL-C and LDL-C exhibited weak but significant correlations with the FIB-4 index (all *p* < 0.05, Appendix A). Furthermore, significant positive correlations were observed between serum levels of FBG, 1hPG, and 2hPG with TG levels (all *p* < 0.001, Appendix A). Conversely, significant inverse correlations were noted between serum levels of FBG, 1hPG, and 2hPG and HDL-C levels (all *p* < 0.001, Appendix A).

### 3.2. Clinical Characteristics of MASLD Patients Subgrouped by 1hPG and 2hPG

Subjects with normal 1hPG and 2hPG levels accounted for the highest proportion of non-MASLD patients (45.5%). However, patients with elevated levels of both 1hPG and 2hPG were common in MASLD patients (39.6%), both in MASLD patients with normal (31.4%) and abnormal (69.0%) fasting glucose (Figure 1A). MASLD patients in the subgroup with normal 1hPG and 2hPG levels were younger than those in the other three subgroups (Table 2). Additionally, MASLD patients with elevated 2hPG levels, regardless of 1hPG status, exhibited higher BMI, WC, WHR, FINS, HOMA-IR, HbA1c, TG, and UA levels, but lower HDL-C levels (all *p* < 0.001). Notably, MASLD patients with elevated levels of both 1hPG and 2hPG had the highest levels of FBG, 1hPG, and 2hPG compared to the other three groups (all *p* < 0.001, Table 2). In addition, there was significant difference in the proportion of moderate–severe steatosis between the four group, with the highest proportion in MASLD patients with elevated 1hPG levels (85.6%) (*p* = 0.02). Those with elevated levels of both 1hPG and 2hPG showed the highest prevalence of liver injury (46.2%) and significant liver fibrosis (27.1%), and those with elevated 2hPG levels had the highest prevalence of atherosclerosis (23.5%) (Figure 1B).

### 3.3. Associations of Glucose Status with Clinical Outcomes

Logistic regression analyses were conducted to investigate the relationships between glucose status and the risk of clinical outcomes in MASLD patients (Table 3). Compared to patients with normal 1hPG and 2hPG levels, patients with elevated 1hPG levels were found to have an increased risk of moderate–severe steatosis (OR = 2.19, 95% CI: 1.13–4.25, *p* = 0.02) after adjusting for potential confounders including sex, age, BMI, TG, HDL-C, UA, and ALT. Elevated levels of both 1hPG and 2hPG were identified as a risk factor for liver injury (OR = 2.03, 95% CI: 1.44–2.86, *p* < 0.001). Additionally, MASLD patients with elevated 2hPG levels, regardless of the 1hPG status, exhibited a higher risk of significant liver fibrosis, with ORs of 1.99 (95% CI: 1.15–3.45, *p* < 0.001) and 2.72 (95% CI: 1.79–4.11, *p* < 0.001), respectively. Furthermore, either 1hPG or 2hPG levels were significantly associated with atherosclerosis, and there were significant dose-dependent associations between glucose status and atherosclerosis risk (OR = 2.77, 95% CI: 1.55–4.96, *p* < 0.001 for elevated 1hPG; OR = 2.98, 95% CI = 1.54–5.78, *p* = 0.001 for elevated 2hPG; OR = 2.41, 95% CI = 1.38–4.21, *p* = 0.001 for elevated both 1hPG and 2hPG). Elevated levels of 1hPG, 2hPG, and relative confounders (as shown above) were used to construct a receiver operating characteristic curve (ROC) for predicting moderate–severe steatosis, liver injury, liver fibrosis, and atherosclerosis. For patients with MASLD, the areas under the ROC for predicting steatosis were 0.64 for an elevated 1hPG status and 0.50 for elevated 2hPG (Figure 2A). Regarding liver injury, the areas under the ROC were 0.58 for an elevated 1hPG status and 0.60 for elevated 2hPG (Figure 2B). As for liver fibrosis, the areas under the ROC were 0.58 for an elevated 1hPG status and 0.56 for elevated 2hPG (Figure 2C). Similarly, the areas under the ROC for predicting atherosclerosis were 0.64 for an elevated 1hPG status and 0.62 for elevated 2hPG (Figure 2D).

### 3.4. The Relationship of Glucose Status and Clinical Outcome Among Different Subgroups

We further evaluated the association between glucose status and the risk of clinical outcomes in MASLD patients stratified by FBG status, sex, and age (Figure 3). The clinical baseline characteristics among different subgroups are presented in Appendix A. In male MASLD patients, the impact of the 1hPG and 2hPG status on the prevalence of clinical outcomes was consistent with the overall findings in all MASLD patients (Appendix A). After adjusting for potential factors, elevated 1hPG levels, regardless of 2hPG status, were associated with increased risk of liver injury (all *p* < 0.05). By contrast, elevated 2hPG levels, regardless of 1hPG status, were related to the increased risk of liver fibrosis (all *p* < 0.05). Elevated 2hPG levels were found to be risk factors for atherosclerosis (*p* = 0.008). In female MASLD patients, those with elevated 1hPG levels exhibited the highest prevalence of moderate–severe steatosis (86.0%), liver fibrosis (24.6%), and atherosclerosis (19.2%) (Appendix A). Elevated 1hPG levels were significantly associated with moderate–severe steatosis (*p* = 0.03). Elevated 2hPG levels, regardless of 1hPG status, were linked to the risk of liver injury (all *p* < 0.05). Elevated levels of both 1hPG and 2hPG levels were risk factors for liver fibrosis (*p* = 0.02). Elevated 1hPG levels, regardless of 2hPG status, were linked to atherosclerosis risk (all *p* < 0.05) (Appendix A).

For MASLD patients aged ≥40 years, individuals with elevated 1hPG levels showed the highest prevalence of moderate–severe steatosis (86.1%) and atherosclerosis (27.4%) (Appendix A). Elevated levels of both 1hPG and 2hPG were related to the increased risk of liver injury (*p* = 0.02) and liver fibrosis (*p* = 0.02). Either elevated 1hPG or 2hPG levels were associated with atherosclerosis risk (all *p* < 0.05). In MASLD patients aged <40 years, individuals with elevated 2hPG levels had the highest prevalence of liver injury (64.8%) and atherosclerosis (18.5%) (Appendix A). Elevated 2hPG levels, regardless of 1hPG status, were linked to the increased risk of liver injury (all *p* < 0.05). Either 1hPG or 2hPG levels were associated with the risk of atherosclerosis (all *p* < 0.05) (Appendix A). Among MASLD patients with normal FBG levels, trends in the 1hPG and 2hPG status on the prevalence of clinical outcomes were consistent with those observed in the overall MASLD group (Figure 1C). Elevated 1hPG levels were risk factors for moderate–severe steatosis (*p* = 0.02), while elevated levels of both 1hPG and 2hPG levels were related to liver injury (*p* = 0.002) and liver fibrosis (*p* = 0.001) and elevated 1hPG levels, regardless of 2hPG status, were associated with atherosclerosis risk (all *p* < 0.05). Among MASLD patients with abnormal FBG levels, subjects with elevated 1hPG levels had the highest prevalence of moderate–severe steatosis (87.5%) and liver injury (57.7%). Those with elevated 2hPG levels had the highest prevalence of liver fibrosis (34.5%), while individuals with elevated levels of both 1hPG and 2hPG had the greatest prevalence of atherosclerosis (28.6%) (Figure 1D). Elevated 2hPG levels, regardless of 1hPG status, were associated with liver fibrosis (all *p* < 0.05), while elevated 1hPG levels were related to the risk of atherosclerosis (*p* = 0.001) (Appendix A). The proportion of participants with elevated 1hPG levels was consistently higher than those with elevated 2hPG levels, regardless of insulin resistance status (HOMA-IR <2.5: 26.2% vs. 10.6%, *p* < 0.05; HOMA-IR ≥2.5: 20.3% vs. 16.3%, *p* < 0.05) (Appendix A). Moreover, no significant differences were found in the risk of clinical outcomes, including moderate–severe steatosis, liver injury, liver fibrosis, and atherosclerosis, between the elevated 1hPG and 2hPG groups, regardless of insulin resistance status (Appendix A). Similarly, the proportion of elevated 1hPG levels remained higher than the 2hPG levels, irrespective of HOMA-β status (HOMA-β < 100: 24.5% vs. 9.5%, *p* < 0.05; HOMA-β ≥ 100: 24.1% vs. 14.5%, *p* < 0.05) (Appendix A), with no significant differences in clinical outcomes between the elevated 1hPG and 2hPG groups (Appendix A).

## 4. Discussion

This retrospective cohort study of MASLD patients demonstrated a distinct association between blood glucose status at 1 and 2 h following the OGTT and the risk of various clinical outcomes, including moderate to severe hepatic steatosis, liver injury, liver fibrosis, and atherosclerosis. Elevated 1hPG levels were significantly correlated with the degree of hepatic steatosis. The simultaneous increase in both 1hPG and 2hPG levels was linked to liver injury. Elevated 2hPG levels, irrespective of 1hPG status, were associated with liver fibrosis. Furthermore, both the 2hPG and 1hPG status were related to the risk of atherosclerosis. These findings suggest that persistent postprandial elevation of blood glucose levels may be involved in the development of MASLD.

The relationship between impaired glucometabolic regulation and MASLD has emerged as a significant research focus [24,25]. Recent investigations involving 656 patients with morbid obesity without diagnosed diabetes have indicated that 1hPG levels ≥8.6 mmol/L may be effective in identifying a more severe metabolic disturbance, characterized by elevated glycemic levels, reduced insulin sensitivity, and significantly impaired β-cell function [26]. Our current study further reveals a positive correlation between high 1hPG levels and the risk of moderate–severe steatosis and atherosclerosis in patients with MASLD, even among those with normal FBG levels (<5.6 mmol/L). Consistent with our findings, a cross-sectional study involving 710 Caucasian participants showed that 1hPG levels ≥8.6 mmol/L were associated with a higher prevalence of fatty liver via transient elastography [9]. Another prospective cohort study with a 39-year follow-up showed a significant relationship of 1-h glucose levels with vascular complications and mortality [27]. Moreover, serum 1hPG was identified as an independent risk factor for a proatherogenic risk profile and various cardiovascular risk factors, including thrombosis, endothelial dysfunction, oxidative stress, unfavorable lipid profiles, elevated blood pressure, inflammatory markers, and uric acid levels [28,29,30]. Furthermore, 1hPG was associated with increased arterial stiffness, heightened carotid intima–media thickness, augmented left ventricular mass, and impaired left ventricular diastolic function [28]. Thus, it is vital to manage serum 1hPG levels in patients with MASLD to prevent adverse outcomes, especially in patients who are overlooked due to normal fasting glucose readings.

The National Diabetes Data Group introduced the term “impaired glucose tolerance (IGT)” in 1979, defined as 2hPG levels ranging from 7.8 to 11.1 mmol/L [31]. Individuals diagnosed with IGT are at increased risk of developing T2DM, with annual progression rates ranging from 5% to 11% [18]. According to the 2024 joint EASL-EASD-EASO consensus statement, a 2hPG level ≥ 7.8 mmol/L is also recognized as a cardiometabolic risk factor [12]. The study demonstrated that abnormal 2hPG levels are positively correlated with an increased risk of liver fibrosis and atherosclerosis. This correlation is supported by findings from several prospective studies, which indicate that serum 2hPG is a more reliable predictor of atherosclerosis cardiovascular disease and mortality risk [32,33,34,35]. Several studies have indicated that significant fluctuations in plasma glucose levels may have more detrimental effects on the cardiovascular system than sustained elevations in glucose levels. An observational cohort study involving 654 patients with T2DM demonstrated that glycemic variability is linked to an increased risk of developing both micro- and macrovascular complications, such as retinopathy and renal outcomes, myocardial infarctions, strokes, and cardiovascular deaths [36]. Another study with 65 non-diabetes mellitus participants with coronary artery disease also found that a high degree of glucose level fluctuations is associated with cardiovascular events, including myocardial infarction, unstable angina, revascularizations, and cardiovascular death [37]. Furthermore, Marfella et al. found that maintaining plasma glucose levels at 15 mmol/L for 2 h resulted in increases in average heart rate and systolic and diastolic blood pressure, as well as in elevated plasma catecholamine concentrations [38]. At the cellular level, acute hyperglycemia triggers oxidative stress due to excessive generation of free radicals in the mitochondria, leading to apoptotic effects. Moreover, monocyte adhesion to endothelial cells, an early marker of atherosclerosis, increases in response to acute elevations in glucose levels; this effect is particularly pronounced in diabetic rats experiencing postprandial glucose spikes compared to those fed ad libitum, despite the latter group exhibiting higher HbA1c levels [39].

An elevated level of 1hPG indicated a significant degree of beta-cell dysfunction in individuals. This dysfunction paralleled the underlying pathophysiological changes that occur as type 2 diabetes progresses. Specifically, it highlighted the point at which beta-cell insulin secretion becomes inadequate, failing to meet the increased insulin demand necessary to maintain glucose homeostasis. This transition underscores the critical role that beta-cell capacity plays in the early stages of glucose regulation [40]. Furthermore, research has demonstrated that individuals with isolated abnormal 1hPG levels exhibited a more pronounced deficiency in beta-cell function compared to those with isolated abnormal 2hPG levels. Notably, subjects with isolated abnormal 2hPG levels have shown severe impairment in insulin secretion [41]. Our results suggest that 1hPG levels serve as an earlier indicator of disrupted glucose homeostasis compared to 2hPG levels. This finding is consistent with previous research, which has shown that 1hPG levels are more sensitive to changes in glucose tolerance and insulin secretion than 2hPG levels [26,42]. Severe fluctuations in blood glucose can lead to insulin resistance, which is characterized by a significant reduction in cellular sensitivity to insulin [43]. This condition results in diminished insulin efficacy in promoting glucose uptake and metabolism. Importantly, there is a strong correlation between the degree of insulin resistance and the severity of hepatic fatty degeneration [44]. In this context, 1hPG levels exhibit heightened sensitivity, as they more accurately reflect the extent of hepatic fatty degeneration at an earlier stage, thereby serving as a valuable marker for early assessment and intervention. Additionally, 1hPG levels have been shown to predict the development of diabetes and cardiovascular disease regardless of other glucose metabolism markers [40]. Therefore, these findings underscore the potential utility of 1hPG levels in the early detection of metabolic abnormalities associated with diabetes, which could facilitate earlier interventions and improve clinical outcomes.

A retrospective observational study conducted in Portugal highlighted a significant agreement between 1hPG and 2hPG levels in patients with morbid obesity and without a diabetes diagnosis [26]. However, our research has revealed notable differences in the associations of 1hPG and 2hPG levels with clinical outcomes. Our results indicate a significant association between 2hPG and liver fibrosis. One possible explanation for this finding is that the sensitivity of 2hPG is lower than that of 1hPG in detecting changes in disrupted glucose homeostasis [42]. Consequently, an abnormal 2hPG level may reflect a more severe degree of liver injury, potentially indicating the presence of significant liver fibrosis or other pathological changes [42]. Notably, individuals of Asian descent, despite having a lower overall BMI, exhibit higher rates of diabetes, increased prevalence of abdominal obesity, and greater insulin resistance [45], indicating the need for more vigilant monitoring of blood glucose levels at different time points.

One of the several limitations of this study is that the assessment of hepatic steatosis was conducted using unenhanced abdominal CT, which is not the most accurate diagnostic technique [46]. In addition, this study was conducted at a single center with a limited sample size, which necessitates caution when interpreting its findings, particularly in the subgroup analyses stratified by sex, age, and FBG status. This study also did not account for other potential confounding factors such as smoking, alcohol intake, and caffeine consumption. Additionally, plasma glucose and insulin levels at 30 min after glucose loading were not measured in the OGTT, so the insulinogenic index could not be calculated. Blood glucose levels are influenced by trends in insulin secretion, and the insulin clamp test is considered the gold standard for evaluation. However, this test was not conducted in our study, and further studies are necessary to explore the role and mechanisms of insulin secretion and insulin resistance in the relationship between loaded glucose and MASLD. Finally, the cross-sectional design of this study restricted our analysis to the relationships among prevalent steatosis, liver fibrosis, liver injury, and atherosclerosis, thereby limiting our ability to establish longitudinal connections between glucose levels and clinical outcomes over time. Future cohort studies with larger sample sizes and long follow-ups are required to confirm these results.

## 5. Conclusions

Abnormal glucose regulation observed at various time points during the OGTT has been distinctly associated with an increased risk of adverse clinical outcomes in patients with MALSD, particularly alongside the occurrence of atherosclerosis. These findings underscore the necessity for clinicians to acknowledge that the screening and management of MALSD requires monitoring 1hPG levels.

## Figures and Tables

**Figure 1 nutrients-17-00152-f001:**
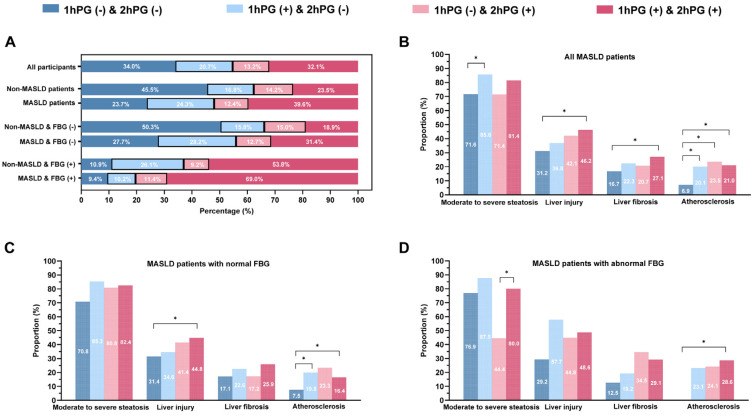
Plot of glucose status and clinical outcomes in MASLD and NAFLD-non-MASLD subjects. OGTT cross groups: 1hPG (−) and 2hPG (−) indicates that 1hPG and 2hPG are both normal. 1hPG (+) and 2hPG (−) indicates that 1hPG is abnormal, while 2hPG is normal. 1hPG (−) and 2hPG (+) indicates that 2hPG is abnormal, while 1hPG is normal. 1hPG (+) and 2hPG (+) indicates that 1hPG and 2hPG are both abnormal. Abbreviation: FBG, fasting blood glucose; 1hPG, 1-h post-load plasma glucose; 2hPG, 2-h post-load plasma glucose; MASLD, metabolic dysfunction-associated steatotic liver disease. (**A**) Distribution of blood glucose status in MASLD. (**B**) Prevalence of clinical outcomes in MASLD. (**C**) Prevalence of clinical outcomes in MASLD subjects among FBG subgroup (**D**) Prevalence of clinical outcomes in MASLD subjects among FBG subgroup. * *p* < 0.05.

**Figure 2 nutrients-17-00152-f002:**
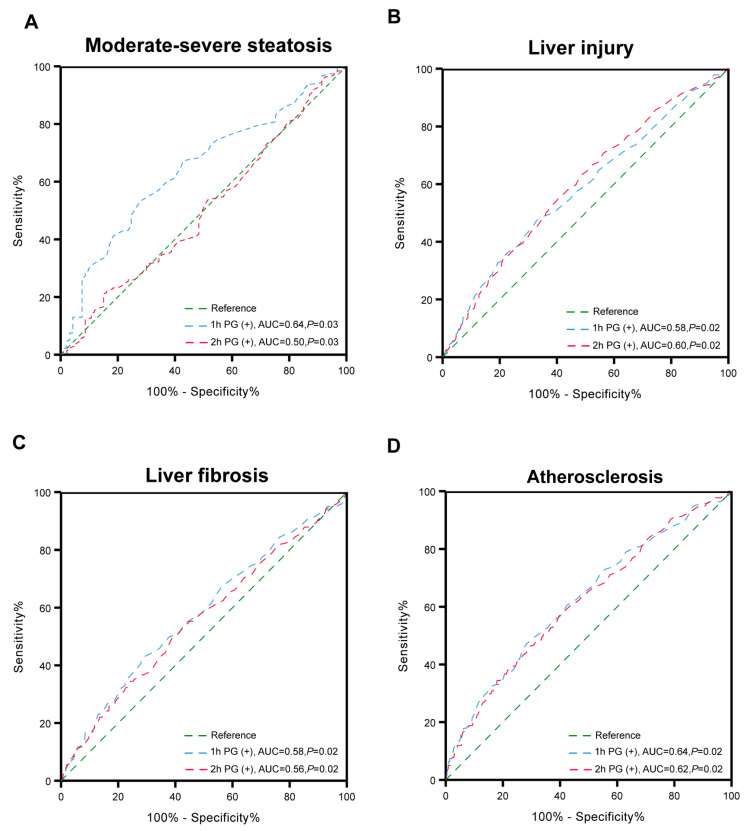
Receiver operator characteristic (ROC) curve predicting moderate–severe steatosis, liver injury, liver fibrosis, and atherosclerosis with glucose status. Abbreviation: 1hPG, 1-h post-load plasma glucose; 2hPG, 2-h post-load plasma glucose. (**A**) Receiver operator characteristic (ROC) curve predicting moderate-severe steatosis with 1hPG (+) or 2hPG (+). (**B**) Receiver operator characteristic (ROC) curve predicting liver injury with 1hPG (+) or 2hPG (+). (**C**) Receiver operator characteristic (ROC) curve predicting liber fibrosis with 1hPG (+) or 2hPG (+). (**D**) Receiver operator characteristic (ROC) curve predicting atherosclerosis with 1hPG (+) or 2hPG (+).

**Figure 3 nutrients-17-00152-f003:**
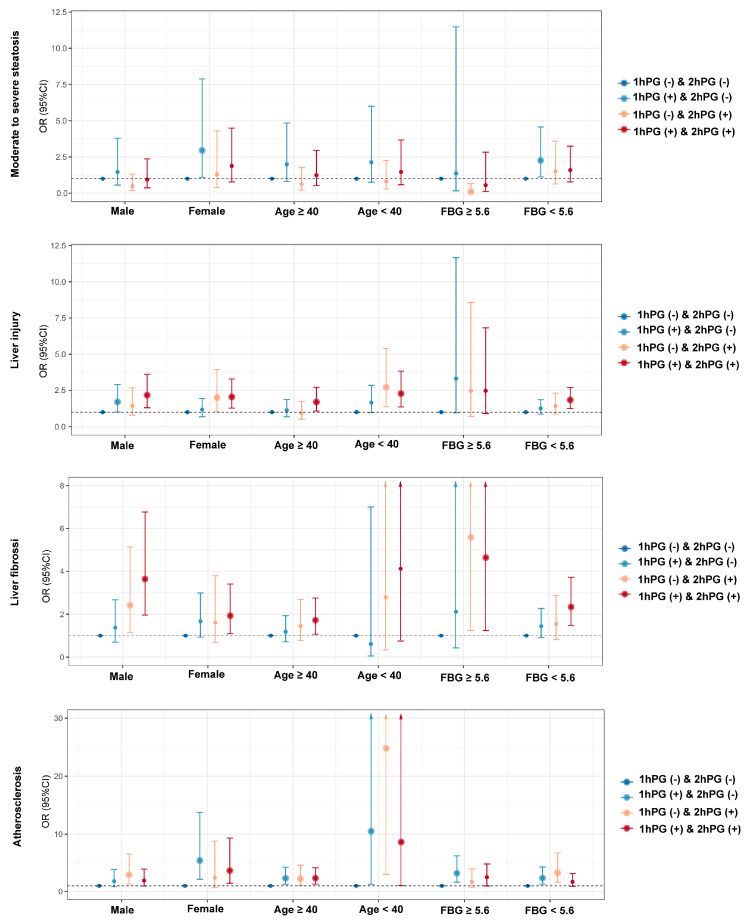
Subgroup analysis of the associations between glucose status and risks of clinical outcomes. Multivariate logistic regressions adjusted for sex, age, BMI, triglycerides, HDL-C, uric acid, and ALT (ALT except for liver injury; age and ALT except for liver fibrosis).

**Table 1 nutrients-17-00152-t001:** Clinical baseline characteristics of all subjects.

Characteristic	Total (*n* = 2214)	Non-MASLD (*n* = 1049)	MASLD (*n* = 1165)	*p*
Age (years)	45.66 ± 15.21	45.54 ± 15.84	45.76 ± 14.62	0.74
Male, *n* (%)	1048 (47.3%)	440 (41.9%)	608 (52.2%)	<0.001
BMI (kg/m^2^)	25.14 ± 4.91	23.18 ± 3.84	26.90 ± 5.09	<0.001
WC (cm)	88.38 ± 12.18	83.91 ± 10.22	92.40 ± 12.41	<0.001
WHR	0.91 ± 0.07	0.89 ± 0.07	0.92 ± 0.06	<0.001
Hypertension, *n* (%)	1025 (46.3%)	396 (37.8%)	629 (54.0%)	<0.001
T2DM, *n* (%)	384 (17.3%)	118 (11.2%)	266 (19.4%)	<0.001
FBG (mmol/L)	4.80 (4.50, 5.30)	4.70 (4.40, 5.10)	5.00 (4.60, 5.50)	<0.001
1hPG (mmol/L)	8.70 (7.70, 10.00)	8.30 (7.30, 9.40)	9.10 (8.00, 10.70)	<0.001
2hPG (mmol/L)	7.50 (6.20, 9.70)	7.00 (5.80, 8.70)	7.90 (6.60, 10.60)	<0.001
FINS (μU/mL)	7.44 (4.79, 10.42)	5.74 (4.04, 8.83)	8.98 (5.75, 12.31)	<0.001
HOMA-IR	1.61 (1.00, 2.37)	1.22 (0.82, 1.87)	2.00 (1.25, 2.81)	<0.001
HOMA-β (%)	105.7 (69.23, 164.95)	95.87 (63.64, 149.67)	113.20 (75.27, 178.46)	<0.001
HBA1c (%)	5.73 (5.30, 5.90)	5.73 (5.30, 5.86)	5.73 (5.36, 5.90)	0.048
CHOL (mmol/L)	4.93 ± 1.18	4.80 ± 1.19	5.05 ± 1.17	<0.001
TG (mmol/L)	1.32 (0.95, 1.84)	1.12 (0.83, 1.59)	1.50 (1.11, 2.17)	<0.001
HDL-C (mmol/L)	1.22 ± 0.34	1.27 ± 0.36	1.17 ± 0.31	<0.001
LDL-C (mmol/L)	3.08 ± 0.85	2.96 ± 0.84	3.20 ± 0.85	<0.001
UA (μmol/L)	382.71 ± 114.08	354.09 ± 104.41	408.48 ± 116.29	<0.001
ALT (U/L)	21.00 (14.00, 32.00)	17.00 (12.00, 26.00)	25.00 (17.00, 39.00)	<0.001
AST (U/L)	21.00 (17.00, 27.00)	20.00 (17.00, 25.00)	22.00 (18.00, 29.00)	<0.001
ALP (U/L)	72.00 (61.00, 87.00)	71.00 (60.00, 86.00)	74.00 (63.00, 88.00)	0.007
ALB(g/L)	40.90 (38.30, 43.40)	40.20 (37.70, 43.00)	41.30 (39.00, 43.90)	<0.001
Platelet (10^9^/L)	252.03 ± 70.74	247.99 ± 70.16	255.66 ± 71.09	0.10
FIB-4 index	0.85 (0.57, 1.27)	0.82 (0.53, 1.20)	0.88 (0.58, 1.30)	0.004
CACS category				<0.001
0, *n* (%)	1208 (54.6%)	665 (63.4%)	543 (46.6%)	
1–99, *n* (%)	758 (34.2%)	339 (32.3%)	419 (36.0%)	
100–399, *n* (%)	248 (11.2%)	43 (4.3%)	183 (17.4%)	
>400, *n* (%)	0 (0.0%)	2 (0.2%)	20 (1.7%)	

Abbreviation: BMI, body mass index; WC, waist circumference; WHR, waist-to-hip ratio; T2DM, diabetes mellitus type 2; FBG, fasting blood glucose; 1hPG, 1-h post-load plasma glucose; 2hPG, 2-h post-load plasma glucose; FINS, fasting serum insulin; HOMA-IR, homeostasis model assessment of insulin resistance; HOMA-β, homeostasis model assessment of β-cell function; HBA1c, glycosylated hemoglobin; CHOL, cholesterol; TG, triglyceride; HDL-C, high-density lipoprotein cholesterol; LDL-C, low-density lipoprotein cholesterol; UA, uric acid; ALT, alanine aminotransferase; AST, aspartate transaminase; ALB, albumin; ALP, alkaline phosphatase; FIB-4, fibrosis 4 index; CACS, coronary artery calcium score.

**Table 2 nutrients-17-00152-t002:** Clinical characteristics of MASLD patients grouped by 1h and 2h PG status.

Characteristics	1hPG (−) and 2hPG (−) (*n* = 276)	1hPG (+) and 2hPG (−) (*n* = 283)	1hPG (−) and 2hPG (+) (*n* = 145)	1hPG (+) and 2hPG (+) (*n* = 461)	*p*
Age (years)	41.31 ± 15.68	45.54 ± 15.15 **a	46.06 ± 14.61 *a	48.47 ± 12.91 ***a	<0.001
Male, *n* (%)	116 (42.0%)	153 (54.1%) *a	91 (62.8%) *a	248 (53.8%) *a	<0.001
BMI (kg/m^2^)	25.64 ± 5.50	25.88 ± 5.85	28.10 ± 4.26 ***a ***b	27.88 ± 4.33 ***a ***b	<0.001
WC (cm)	89.29 ± 13.96	90.40 ± 13.51	95.40 ± 10.71 ***a ***b	94.53 ± 10.46 ***a ***b	<0.001
WHR	0.91 ± 0.14	0.91 ± 0.06	0.93 ± 0.06 ***a **b	0.94 ± 0.11 ***a ***b	<0.001
Hypertension, *n* (%)	118 (42.8%)	118 (41.7%)	93 (64.1%) *a *b	300 (65.1%) *a *b	<0.001
T2DM, *n* (%)	0 (0.0%)	0 (0.0%)	34 (23.5%) ***a ***b	232 (50.3%) ***a ***b **c	<0.001
FBG (mmol/L)	4.60 (4.30, 4.90)	4.90 (4.60, 5.30) ***a	4.90 (4.60, 5.40) ***a	5.30 (4.90, 5.90) ***a ***b ***c	<0.001
1hPG (mmol/L)	7.50 (6.70, 8.00)	9.40 (9.00, 10.65) ***a	7.90 (7.50, 8.20) ***b	10.50 (9.40, 13.00) ***a ***b ***c	<0.001
2hPG (mmol/L)	6.40 (5.50, 7.10)	6.80 (6.10, 7.20)	9.60 (8.50, 10.90) ***a ***b	11.00 (9.10, 13.40) ***a ***b *c	<0.001
FINS (μU/mL)	7.47 (4.59, 11.23)	8.65 (5.07, 11.91)	9.53 (6.92, 15.35) ***a ***b	8.98 (7.15, 13.00) ***a ***b	<0.001
HOMA-IR	1.60 (0.94, 2.26)	1.83 (1.06, 2.63)	2.16 (1.45, 3.17) ***a **b	2.27 (1.68, 3.19) ***a ***b	<0.001
HOMA-β (%)	94.53 (65.12, 146.34)	105.80 (68.67, 167.76)	106.36 (69.72, 171.16)	113.33 (74.00, 179.57) ***a	<0.001
HBA1c (%)	5.50 (5.20, 5.73)	5.50 (5.30, 5.77)	5.73 (5.50, 6.10) ***a ***b	5.86 (5.60, 6.30) ***a ***b	<0.001
CHOL (mmol/L)	5.03 ± 1.12	4.99 ± 1.09	5.13 ± 1.25	5.08 ± 1.22	0.60
TG (mmol/L)	1.27 (0.95, 1.69)	1.32 (0.98, 1.86)	1.87 (1.31, 2.56) ***a ***b	1.66 (1.23, 2.38) ***a ***b	<0.001
HDL-C (mmol/L)	1.26 ± 0.34	1.21 ± 0.32	1.06 ± 0.26 ***a ***b	1.12 ± 0.27 ***a ***b	<0.001
LDL-C (mmol/L)	3.16 ± 0.80	3.16 ± 0.86	3.25 ± 0.83	3.23 ± 0.88	0.45
UA (μmol/L)	385.22 ± 115.29	396.71 ± 117.97	440.12 ± 121.55 ***a **b	419.68 ± 110.76 ***a *b	<0.001
ALT (U/L)	21.00 (14.00, 32.00)	23.00 (17.00, 38.50) *a	27.00 (18.00, 40.00) **a	28.00 (20.00, 41.00) ***a *b	<0.001
AST (U/L)	21.00 (17.00, 25.00)	22.00 (18.00, 29.00) *a	22.00 (18.00, 31.00)	24.00 (19.00, 31.00) ***a	<0.001
ALP (U/L)	74.00 (60.00, 88.00)	72.00 (62.00, 86.00)	74.00 (64.00, 87.00)	74.00 (63.00, 89.00)	0.40
ALB (g/L)	41.10 (38.60, 43.42)	41.50 (39.10, 43.65)	41.60 (38.80, 44.00)	41.20 (39.00, 44.00)	0.61
CT value (HU)	33.00 (31.00, 41.50)	32.50 (28.62, 35.25)	32.50 (30.00, 42.25)	32.25 (24.00, 34.38)	0.11
Platelet (10^9^/L)	253.23 ± 70.64	253.23 ± 70.64	253.23 ± 70.64	253.23 ± 70.64	0.26
FIB-4 index	0.73 (0.46, 1.06)	0.85 (0.56, 1.23) *a	0.79 (0.49, 1.09)	0.87 (0.61, 1.32) ***a *c	<0.001
CACS category > 100, *n* (%)	19 (6.9%)	57 (20.1%) *a	34 (23.5%) *a	97 (21.0%) *a	<0.001

Abbreviation: BMI, body mass index; WC, waist circumference; WHR, waist-to-hip ratio; T2DM, diabetes mellitus type 2; FBG, fasting blood glucose; 1hPG, 1-h post-load plasma glucose; 2hPG, 2-h post-load plasma glucose; FINS, fasting serum insulin; HOMA-IR, homeostasis model assessment of insulin resistance; HOMA-β, homeostasis model assessment of β-cell function; HBA1c, glycosylated hemoglobin; CHOL, cholesterol; TG, triglyceride; HDL-C, high-density lipoprotein cholesterol; LDL-C, low-density lipoprotein cholesterol; UA, uric acid; ALT, alanine aminotransferase; AST, aspartate transaminase; ALP, alkaline phosphatase; ALB, albumin; FIB-4 index, fibrosis-4 index; CACS, coronary artery calcium score. OGTT cross groups: 1hPG (−) and 2hPG (−) indicates that 1hPG and 2hPG are both normal; 1hPG (+) and 2hPG (−) indicates that 1hPG is abnormal, while 2hPG is normal; 1hPG (−) and 2hPG (+) indicates that 2hPG is abnormal, while 1hPG is normal; 1hPG (+) and 2hPG (+) indicates that 1hPG and 2hPG are both abnormal. a, compared with 1hPG (−) and 2hPG (−); b, compared with 1hPG (+) and 2hPG (−); c, compared with 1hPG (−) and 2hPG (+). * *p* < 0.05, ** *p* < 0.01, *** *p* < 0.001.

**Table 3 nutrients-17-00152-t003:** Logistic regression analysis of the relationships between glucose status and clinical outcomes.

Characteristics	1hPG (−) and 2hPG (−) (*n* = 276)	1hPG (+) and 2hPG (−) (*n* = 283)	1hPG (−) and 2hPG (+) (*n* = 145)	1hPG (+) and 2hPG (+) (*n* = 461)	*p* for Trend
Moderate–severe steatosis					
Crude	Reference	**2.36 (1.25–4.48)**	0.99 (0.51–1.93)	1.74 (0.99–3.08)	0.28
Model 1	Reference	**2.12 (1.11–4.07)**	0.84 (0.42–1.67)	1.47 (0.81–2.68)	0.63
Model 2	Reference	**2.19 (1.13–4.25)**	0.80 (0.39–1.63)	1.44 (0.78–2.64)	0.72
Liver injury					
Crude	Reference	1.28 (0.90–1.82)	**1.60 (1.06–2.43)**	**1.90 (1.39–2.60)**	<0.001
Model 1	Reference	1.42 (0.99–2.04)	**1.67 (1.08–2.58)**	**2.13 (1.52–2.99)**	<0.001
Model 2	Reference	1.40 (0.97–2.02)	1.55 (0.99–2.43)	**2.03 (1.44–2.86)**	<0.001
Liver fibrosis					
Crude	Reference	1.43 (0.94–2.18)	1.30 (0.78–2.18)	**1.86 (1.28–2.71)**	0.002
Model 1	Reference	1.44 (0.93–2.22)	1.69 (0.99–2.89)	**2.43 (1.63–3.63)**	<0.001
Model 2	Reference	1.49 (0.96–2.32)	**1.99 (1.15–3.45)**	**2.72 (1.79–4.11)**	<0.001
CVD					
Crude	Reference	**3.41 (1.97–5.91)**	**4.14 (2.26–7.58)**	**3.60 (2.15–6.04)**	<0.001
Model 1	Reference	**2.73 (1.53–4.88)**	**2.85 (1.49–5.44)**	**2.31 (1.34–4.01)**	0.049
Model 2	Reference	**2.77 (1.55–4.96)**	**2.98 (1.54–5.78)**	**2.41 (1.38–4.21)**	0.04

Model 1 adjusted for sex, age, and BMI (age except for liver fibrosis). Model 2 adjusted for sex, age, and BMI, triglycerides, HDL-C, uric acid, ALT (ALT except for liver injury, age and ALT except for liver fibrosis). Highlights refer to odds ratios with significant differences (*p* < 0.05).

## Data Availability

The data presented in this study are available upon reasonable request from the corresponding author. The data are not publicly available due to privacy reasons.

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
