# Peer review of "Comparisons of Post-Load Glucose at Different Time Points for Identifying High Risks of MASLD Progression"

_nutrients, 2024, doi:10.3390/nu17010152_

Round 1

Reviewer 1 Report

Comments and Suggestions for Authors

Reviewer’s comments

This original article written by Ten and Luo et al. primarily focuses on a clinical significance in 1hPG and 2hPG at oGTT. This original article has its own originality and seems to be very interesting. I appreciate authors’ efforts. However, the explanation for the methods in this study is insufficient or ambiguous in several parts. The interpretation of the results obtained from this study is also insufficient. Several revisions are required for the better quality of this review article. Please refer to the comments shown below.

Major

#1. Steatotic liver disease subclassification has been recently conducted. The term, “NAFLD” is shifted to “MASLD”. Therefore, the reference on the concept of MASLD should be cited.

#2. CVD was identified through CT angiography in this study. However, the cutoff value for the score was not noted at all. The data on the CT angiography were missing in Table 1and Table 2.

#3. CVD patients were diagnosed through the score by CT angiography. Patients with heart attack were excluded in this study. However, CVD patients usually cause heart attack. Therefore, the patients enrolled in this study did not meet the criteria for CVD. The term “CVD group” should be altered to “atherosclerosis group”.

#4. (age, except for liver fibrosis: Page 7, line 257) may be wrong. The term should be corrected to “Age, ALT, and AST except for liver fibrosis”? Similarly, (ALT except for liver injury: Page 7, line 258) may be corrected to (ALT and AST except for liver injury). (age, and ALT except for liver fibrosis: Page 7, line 259) may be corrected to (age, ALT, and AST except for liver fibrosis.

#5. In Table 2, there was not a significant difference in CT values between 1hPG(+) and 2hPG(-) group and 1hPG(+) and 2hPG(+) group. However, why the elevated 1hPG were significantly correlated with the degree of hepatic steatosis (Page 10, lines 312-313)? The authors should describe a putative mechanism.

#6. This study revealed that elevated 2hPG levels were associate with liver fibrosis irrespective of 1hPG (Page 10, line 314). However, FIB4-index in the group of 1hPG(-) and 2hPG(+) was significantly lower than that in the group of 1hPG(+) and 2hPG (+) (Table 2). The authors should also describe the reason.

#7. In Table 2, the mean 2h-PG level in the group of 1hPG(+) and 2gPG(+) was 11.00 mmol/L (9.40, 13.00),indicating that some patients were diagnosed with type 2 DM. How many patients were diagnosed with type 2 DM through oGTT?

#8. At the end of the manuscript, the paragraph for the conclusion in this study should be inserted.

Minor

#1. Fibrosis was selected as one of the keywords. “Fibrosis” should be corrected to “Liver fibrosis”

Likewise, fibrosis (Page 2, line 70) should be corrected to liver fibrosis. Aminopherase was also selected as the keyword (Page 1, line 42). What was it?

#2. “Kidney failure” (Page 3, line 107) should be corrected to “renal failure”.

#3. Figure 2 was after Figure 3 in the text. The order was backward.

Figure 1Figure 3Figure 2 should be corrected to Figure 1Figure 2Figure 3.

#4. HOMA-IR is an abbreviation for “homeostasis model assessment of insulin resistance” Homeostasis model assessment of insulin resistance index (Page 5, lines 188-189) should be corrected to homeostasis model assessment of insulin resistance.

#5. The spell out for FIB-4 index (Page 6, line 223) was not required.

#6. The spell out for ALT (Page 8, line 265) was wrong.

#7. The statement on “Data Availability” should be described at the end of the manuscript.

Comments on the Quality of English Language

The quality of English language can be improved in several parts.

Author Response

Reviewer #1:

This original article written by Teng and Luo et al. primarily focuses on a clinical significance in 1hPG and 2hPG at OGTT. This original article has its own originality and seems to be very interesting. I appreciate authors’ efforts. However, the explanation for the methods in this study is insufficient or ambiguous in several parts. The interpretation of the results obtained from this study is also insufficient. Several revisions are required for the better quality of this review article. Please refer to the comments shown below.

Major

#1. Steatotic liver disease subclassification has been recently conducted. The term, “NAFLD” is shifted to “MASLD”. Therefore, the reference on the concept of MASLD should be cited.

Reply 1: Thank you very much for your positive feedback and interest in our study. Based on your suggestion, we have added the description that “MASLD was renamed from non-alcoholic fatty liver disease (NAFLD) in 2023, and several studies have reported that approximately 99% of patients with NAFLD meet the MASLD criteria.” (see Page 2, lines 48-51). And we have added the relevant references to the paper (see Page 13, lines 444-447).

Reference 2:  Rinella ME, Lazarus JV, Ratziu V,et al. A multisociety Delphi consensus statement on new fatty liver disease nomenclature. J Hepatol. 2023;79:1542-1556.

Reference 3:  Hagström H, Vessby J, Ekstedt M, Shang Y. 99% of patients with NAFLD meet MASLD criteria and natural history is therefore identical. J Hepatol. 2024;80:e76-e77.

Changes in the text: Page 2, lines 48-51; Page 13, lines 444-447; References 2-3.

#2. CVD was identified through CT angiography in this study. However, the cutoff value for the score was not noted at all. The data on the CT angiography were missing in Table 1 and Table 2.

Reply 2: We feel very sorry for our not clear expressions. We have now included the cutoff values for the coronary artery calcium (CAC) score in the text: “Clinical events related to atherosclerosis were identified using CT angiography, which employed the coronary artery calcium (CAC) score. Furthermore, CAC severity was categorized into four groups according to scores of 0, 0–100, 100-399, and > 399 representing as stage normal, mild, moderate, and severe CAD risk. Additionally, the threshold for inclusion in the atherosclerosis group was set at a CAC score of 100 or above” (see Page 4, lines 163-168). Additionally, we have included the data on CT angiography in Tables 1 and 2.

Changes in the text: Page 4, lines 163-168; Revised Tables 1-2.

#3. CVD patients were diagnosed through the score by CT angiography. Patients with heart attack were excluded in this study. However, CVD patients usually cause heart attack. Therefore, the patients enrolled in this study did not meet the criteria for CVD. The term “CVD group” should be altered to “atherosclerosis group”.

Reply 3: Thanks for the constructive suggestions. We have excluded recent heart attack patients to focus specifically on atherosclerosis. As you suggested, we have replaced the term “CVD group” with “atherosclerosis group” for enhanced clarity. The necessary modifications have been made accordingly.

Changes in the text: Page 1, lines 34, 35, 38, and 43; Page 2, lines 57, 81, and 85; Page 4, lines 163, 167, and 177-178; Page 5, line 215; Page 7, lines 249, 250, 254, and 260; Page 9, lines 270, 281, 283, and 287-288; Page 10, lines 295, 298, 300, and 302; Page 11, lines 308, 312, 314, 319-320, 326, 330, 338, 358, and 360; Page 12, lines 375; Page 13, lines 415-416, and 423; Revised Figures 1-3; Revised Figures S1-S2; Revised Table 3; Revised Tables S7-S9.

#4. (age, except for liver fibrosis: Page 7, line 257) may be wrong. The term should be corrected to “Age, ALT, and AST except for liver fibrosis”? Similarly, (ALT except for liver injury: Page 7, line 258) may be corrected to (ALT and AST except for liver injury). (age, and ALT except for liver fibrosis: Page 7, line 259) may be corrected to (age, ALT, and AST except for liver fibrosis.

Reply 4: Thank you for your kind reminding. We appreciate your interest in our logistic regression analysis and subgroup analysis methods. Since ALT and AST are collinear, we included ALT but not AST in our multivariate analysis. In multivariate logistic model 1, we adjusted for sex, age, and BMI for the entire population cohort. In multivariate model 2, we included the variables from model 1, along with triglycerides, HDL-C, uric acid, and ALT.

The FIB-4 scores were calculated using a formula that incorporated age, ALT, and AST. Therefore, when assessing liver fibrosis as an outcome, we excluded age in multivariate model 1 and both excluded age and ALT in multivariate model 2. Similarly, liver injury is evaluated by ALT. When considering liver injury as a logistic outcome, we excluded ALT in the multivariate logistic model 2.

Therefore, “age, except for liver fibrosis” has been revised to “age except for liver fibrosis”; and “age, and ALT except for liver fibrosis” has been revised to “age and ALT except for liver fibrosis” (see Page 8, lines 264-266; Page 10, lines 291-292). Thank you again for your advice.

Changes in the text: Page 8, lines 264-266; Page 10, lines 291-292; Revised Figure  3; Revised Table 3; Revised Tables S7-S9.

#5. In Table 2, there was not a significant difference in CT values between 1hPG(+) and 2hPG(-) group and 1hPG(+) and 2hPG(+) group. However, why the elevated 1hPG were significantly correlated with the degree of hepatic steatosis (Page 10, lines 312-313)? The authors should describe a putative mechanism.

Reply 5: Thank you for your valuable suggestion. In response to your comment, we would like to clarify that the CT value in Table 2 was presented as a continuous variable which doesn’t follow a normal distribution. Non-normal nonparametric comparisons can potentially diminish sensitivity and be susceptible to the influence of extreme values. However, in our logistic regression analysis, we categorized steatosis into moderate and severe groups to more effectively illustrate the proportions of each category. We recognize the potential presence of confounding factors and have taken this into account. As demonstrated in Table 3, we adjusted for these confounding factors to accurately represent the internal relationships between the variables. Furthermore, the statistical power may be limited due to the small sample size, suggesting that the findings should be interpreted with caution (see Page 12, lines 406-407).

Changes in the text: Page 12, lines 406-407.

#6. This study revealed that elevated 2hPG levels were associate with liver fibrosis irrespective of 1hPG (Page 10, line 314). However, FIB4-index in the group of 1hPG(-) and 2hPG(+) was significantly lower than that in the group of 1hPG(+) and 2hPG (+) (Table 2). The authors should also describe the reason.

Reply 6: Thank you for your valuable advice and your interest in the data presented in Table 2. Although the FIB-4 scores of the 1hPG(+) and 2hPG(+) groups exhibited numerically lower values compared to those of their counterparts, these scores did not adhere to a normal distribution. Notably, the application of non-parametric tests for continuous variables that are not normally distributed may compromise sensitivity in statistical analyses. Furthermore, after adjusting for potential confounders, our results indicated that elevated 2hPG levels were associated with liver fibrosis, suggesting a potential link between 2hPG levels and liver fibrosis. Given the limited sample size, we emphasize the need for further studies with larger sample sizes to confirm these findings (see Page 12, lines 406-407; Page 13, lines 417-418).
Changes in the text: Page 12, lines 406-407; Page 13, lines 417-418.

#7. In Table 2, the mean 2h-PG level in the group of 1hPG(+) and 2gPG(+) was 11.00 mmol/L (9.40, 13.00),indicating that some patients were diagnosed with type 2 DM. How many patients were diagnosed with type 2 DM through OGTT?

Reply 7: Thank you for your insightful suggestion. Our study included a total of 2,214 subjects. Among these, 266 of 1,165 subjects with MASLD met the criteria for type 2 diabetes mellitus (T2DM) as determined by an oral glucose tolerance test (OGTT): FBG ≥7.0mmol/L or 2hPG ≥11.1mmol/L. By contrast, 118 out of 1,049 subjects without MASLD also met the criteria for T2DM through the OGTT. Furthermore, we have incorporated this data into the text and revised Table 1 and Table 2 accordingly. In the Materials and Methods section, we have incorporated the diagnostic criteria for Type 2 Diabetes Mellitus (T2DM) (see Page 3, lines 128-130. Page 4, line 189).

Changes in the text: Page 3, lines 128-130. Page 4, line 189; Revised Table 1; Revised Table 2.

#8. At the end of the manuscript, the paragraph for the conclusion in this study should be inserted.

Reply 8: Thank you for your kind reminding. We sincerely apologize for the oversight in omitting the subheading “Conclusion”. We have now included it in the text (see Page 13, line 420).

Changes in the text: Page 13, line 420.

Minor
#1. Fibrosis was selected as one of the keywords. “Fibrosis” should be corrected to “Liver fibrosis”

Likewise, fibrosis (Page 2, line 70) should be corrected to liver fibrosis. Aminopherase was also selected as the keyword (Page 1, line 42). What was it?

Reply 1: Thank you for the reminder. We have revised the term “fibrosis” to “liver fibrosis” throughout the text. Additionally, we are so sorry for the error; the keyword “aminotransferase” has been corrected to “alanine aminotransferase”.

Changes in the text: Page 1, lines 26, 37, 43, and 44; Page 2, lines 77 and 80; Page 10, lines 297 and 307; Page 11, line 313; Page 13, line 415.

#2. “Kidney failure” (Page 3, line 107) should be corrected to “renal failure”.

Reply 2: Thank you for your kind reminding. We have changed “Kidney failure” to “renal failure” (see Page 2, line 56; Page 3, line 113).

Changes in the text: Page 2, line 56; Page 3, line 113.

#3. Figure 2 was after Figure 3 in the text. The order was backward.

Figure 1→Figure 3→Figure 2 should be corrected to Figure 1→Figure 2→Figure3.

Reply 3: We are so sorry for the error. The order of the figures has been adjusted according to your suggestion, and the legends have been reordered accordingly.

Changes in the text: Revised Figure 2, Revised Figure 3; Page 7, lines 256-261; Page 9, lines 269-271, and 274; Page 10, lines 290-292;

#4. HOMA-IR is an abbreviation for “homeostasis model assessment of insulin resistance” Homeostasis model assessment of insulin resistance index (Page 5, lines 188-189) should be corrected to homeostasis model assessment of insulin resistance.

Reply 4: Thanks for your reminding. We have corrected “Homeostasis model assessment of insulin resistance index” into “homeostasis model assessment of insulin resistance”.

Changes in the text: Page 3, line 138; Page 5, lines 196-197; Page 7, lines 226-227; Tables 1-2; Tables S1-S6.

#5. The spell out for FIB-4 index (Page 6, line 223) was not required.

Reply 5: Thank you for your important suggestion. We have removed the spell out for FIB-4 index.

Changes in the text: Page 4, line 161.

#6. The spell out for ALT (Page 8, line 265) was wrong.

Reply 6: We are so sorry for the error. We have corrected the spell out for ALT.

Changes in the text: Page 3, line 137.

#7. The statement on “Data Availability” should be described at the end of the manuscript.

Reply 7: Thank you for your kind reminding. We have added the “Data Availability” in the text.

Changes in the text: Page 13, lines 439-440.

Reviewer 2 Report

Comments and Suggestions for Authors

The work presented by the authors follows a fairly coherent line of reasoning, is relatively well written and derives some very interesting results. It is true that there are a number of limitations associated with the design of the study itself, which the authors admit, but there is still an interesting set of specificities associated with glucose levels one hour (1hPG) and two hours (2hPG) after the start of the oral glucose tolerance test. Based on the analyses carried out, the authors draw the main conclusion that clinicians should read glucose levels at 1hPG in order to properly monitor and manage MASLD.

Some minor comments:

i) The authors should homogeneize the definition of the 1hPG & 2hPG throughut the manuscript. Sometimes it appears together with OGTT other not. If PG means already that it is post OGTT inititation there is no need to create such a confusion and the designations 1hPG and 2hPG hould stand just by themselves.

ii) The authors provide the folloowing sentence in the discussion section: "Recent investigations have shown that1hPG levels of ≥8.6 mmol/L may be useful in identifying individuals with normal glucose tolerance who are at an elevated risk of developing type 2 diabetes mellitus (T2DM), particularly those exhibiting abnormalities in glucose homeostasis, such as reduced insulin sensitivity and β-cell dysfunction [22]". If the individuals are already with abnormalities such as insulin sensitivity and β-cell dysfunction what is the need for an extra 1hPG parameter to infer about the risk for T2DM? Is there a cooperative effct and the risk raises even further?. This needs to be more clearly presented byt he authors.

iii) The sentences "A plausible explanation for this may be that significant fluctuations in plasma glucose levels exert more adverse effects on the cardiovascular system than sustained elevations in glucose levels [34-35]." and "These findings suggested that 1hPG served as an earlier indicator of disrupted glucose homeostasis than 2hPG, emphasizing its potential utility in the early detection of metabolic abnormalities associated with diabetes." need to be further substantiated in the discussion, as they relate to the core of the work itself. 

Author Response

The work presented by the authors follows a fairly coherent line of reasoning, is relatively well written and derives some very interesting results. It is true that there are a number of limitations associated with the design of the study itself, which the authors admit, but there is still an interesting set of specificities associated with glucose levels one hour (1hPG) and two hours (2hPG) after the start of the oral glucose tolerance test. Based on the analyses carried out, the authors draw the main conclusion that clinicians should read glucose levels at 1hPG in order to properly monitor and manage MASLD.

Some minor comments:

  1. i) The authors should homogenize the definition of the 1hPG & 2hPG throughout the manuscript. Sometimes it appears together with OGTT other not. If PG means already that it is post OGTT initiation there is no need to create such a confusion and the designations 1hPG and 2hPG should stand just by themselves.

Reply 1: Thank you for your constructive suggestion. In our study, the terms 1hPG and 2hPG specifically refer to the blood glucose levels measured at 1 hour and 2 hours, respectively, during the OGTT test. We have homogenized the definitions of 1hPG and 2hPG throughout the manuscript to eliminate any potential confusion.

Changes in the text: Page 6, lines 218-220; Page 7, lines 231-233.

  1. ii) The authors provide the following sentence in the discussion section: "Recent investigations have shown that1hPG levels of ≥8.6 mmol/L may be useful in identifying individuals with normal glucose tolerance who are at an elevated risk of developing type 2 diabetes mellitus (T2DM), particularly those exhibiting abnormalities in glucose homeostasis, such as reduced insulin sensitivity and β-cell dysfunction [22]". If the individuals are already with abnormalities such as insulin sensitivity and β-cell dysfunction what is the need for an extra 1hPG parameter to infer about the risk for T2DM? Is there a cooperative effect and the risk raises even further? This needs to be more clearly presented by the authors.

Reply 2: We feel very sorry for our not clear expressions. We stated that there was a problem with the sentence and rewrote it as follows: “Recent investigations involving 656 patients with morbid obesity without diagnosed diabetes have indicated that 1hPG levels ≥8.6 mmol/L may be effective in identifying a more severe metabolic disturbance, characterized by elevated glycemic levels, reduced insulin sensitivity, and significantly impaired β-cell function” (see Page 11, lines 333-337).

Changes in the text: Page 11, lines 333-337.

iii) The sentences "A plausible explanation for this may be that significant fluctuations in plasma glucose levels exert more adverse effects on the cardiovascular system than sustained elevations in glucose levels [34-35]." and "These findings suggested that 1hPG served as an earlier indicator of disrupted glucose homeostasis than 2hPG, emphasizing its potential utility in the early detection of metabolic abnormalities associated with diabetes." need to be further substantiated in the discussion, as they relate to the core of the work itself.

Reply 3: Thank you for your valuable feedback on our manuscript. In response to your suggestions, we have further enhanced the content in the discussion section and included relevant references to support our arguments.

The first sentence has been revised as follows: “Several studies have indicated that significant fluctuations in plasma glucose levels may have more detrimental effects on the cardiovascular system than sustained elevations in glucose levels. An observational cohort study involving 654 patients with T2DM demonstrated that glycemic variability is linked to an increased risk of developing both micro- and macrovascular complications, such as retinopathy and renal outcomes, my-ocardial infarctions, strokes, and cardiovascular deaths [37]. Another study with 65 non-diabetes mellitus participants with coronary artery disease also found that a high degree of glucose level fluctuations is associated with cardiovascular events, including myocardial infarction, unstable angina, revascularizations, and cardiovascular death [38]” (see Pages 11-12, lines 360-369).

The second sentence has been updated to: “Our results suggest that 1hPG levels serve as an earlier indicator of disrupted glucose homeostasis compared to 2hPG levels, which is supported by previous research demonstrating that 1hPG levels are more sensitive to changes in glucose tolerance and insulin secretion than 2hPG levels [43]. Additionally, 1hPG levels have been shown to predict the development of diabetes and cardiovascular disease regardless of other glucose metabolism markers [44]. Therefore, these findings underscore the potential utility of 1hPG levels in the early detection of metabolic abnormalities associated with diabetes, which could facilitate earlier interventions and improve clinical outcomes.” (see Page 12, lines 388-395).

Relevant references:

Reference 37: Cardoso CRL, Leite NC, Moram CBM, et al. Long-term visit-to-visit glycemic variability as predictor of micro- and macrovascular complications in patients with type 2 diabetes: The Rio de Janeiro Type 2 Diabetes Cohort Study. Cardiovasc Diabetol. 2018;17:33.

Reference 38: Akasaka T, Sueta D, Tabata N, et al. Effects of the Mean Amplitude of Glycemic Excursions and Vascular Endothelial Dysfunction on Cardiovascular Events in Nondiabetic Patients With Coronary Artery Disease. J Am Heart Assoc. 2017;6:e004841.

Reference 43: Ferrannini G, De Bacquer D, Gyberg V, et al. Saving time by replacing the standardised two-hour oral glucose tolerance test with a one-hour test: Validation of a new screening algorithm in patients with coronary artery disease from the ESC-EORP EUROASPIRE V registry. Diabetes Res. Clin. Pract. 2022: 183: 109156.

Reference 44: Peng M, He S, Wang J, et al. Efficacy of 1-hour postload plasma glucose as a suitable measurement in predicting type 2 diabetes and diabetes-related complications: A post hoc analysis of the 30-year follow-up of the Da Qing IGT and Diabetes Study. Diabetes Obes Metab. 2024:26: 2329–2338.

Changes in the text: P Pages 11-12, lines 360-369; Page 12, lines 388-395; Page 14, lines 516-520; Page 15, lines 531-536; References 37-38; References 43-44.

Reviewer 3 Report

Comments and Suggestions for Authors

Authors evaluated the predictive value of 1hPG and 2hPG for identifying high risk of MASLD. This study was interesting, and a large number of patients were included. But several issues remined unclear.

1. In the method section, authors described that participants were included prospectively according to EASL2024 criteria. It was impossible to include patients prospectively.

2. The 1hPG and 2hPG were largely affected by calory intake. Authors should clarify which is important whether insulin basal secretion or insulin resistance.  

3. The criteria of CVD was unclear.

4. The 1hPG and 2hPG might be affected by various factors. Multivariate analyses should be performed.

Author Response

Authors evaluated the predictive value of 1hPG and 2hPG for identifying high risk of MASLD. This study was interesting, and a large number of patients were included. But several issues remined unclear.

  1. In the method section, authors described that participants were included prospectively according to EASL2024 criteria. It was impossible to include patients prospectively.

Reply 1: We feel very sorry for our not clear expressions. The study protocol was registered in 2014 and was defined as NAFLD; however, the collected variables encompassed all essential components required for diagnosing MASLD. In the final analysis, the patients included in this study were selected based on the most recent diagnostic criteria for MASLD. Additionally, we have incorporated relevant descriptions in the paper to mitigate any potential confusion (see Pages 2-3, lines 98-100).

Changes in the text: Pages 2-3, lines 98-100.

  1. The 1hPG and 2hPG were largely affected by calory intake. Authors should clarify which is important whether insulin basal secretion or insulin resistance.

Reply 2: Thank you for your valuable comments on our manuscript. We would like to clarify that all 2,214 participants included in our study underwent an Oral Glucose Tolerance Test (OGTT), during which both 1-hour plasma glucose (1hPG) and 2-hour plasma glucose (2hPG) levels were measured. Consequently, calorie intake was standardized throughout this process, enabling us to minimize the influence of varying calorie intake on the 1hPG and 2hPG levels in our analysis.

To address the question of whether insulin basal secretion or insulin resistance is more significant, we added a subgroup analysis based on HOMA-IR status (see Figure S2). Our analysis revealed that the proportion of elevated 1hPG levels was consistently higher than that of elevated 2hPG levels, irrespective of insulin resistance status (the HOMA-IR<2.5 subgroup: 26.2% vs. 10.6%, p<0.05; the HOMA-IR ≥2.5 subgroup: 20.3% vs. 16.3%, p<0.05) (see Figure S2A-B). Furthermore, we found no significant differences in the risk of clinical outcomes, including moderate to severe steatosis, liver injury, liver fibrosis, and atherosclerosis, between the elevated 1hPG and 2hPG groups, irrespective of insulin resistance status (see Figure S2C-D) (see Page 11, lines 315-321). These findings illustrate the association between insulin resistance and loaded blood glucose; however, further studies are necessary to explore the role and mechanisms of insulin secretion and insulin resistance in the relationship between loaded glucose and MASLD (see Page 12, lines 410-414).

Changes in the text:  Page 11, lines 315-321; Page 12, lines 410-414; Revised Figure S2.

  1. The criteria of CVD was unclear.

Reply 3: Thank you for your kind reminding. We have added further details regarding the criteria for cardiovascular disease (CVD) as follows: “Clinical events related to atherosclerosis were identified using CT angiography, which employed the coronary artery calcium (CAC) score. Furthermore, CAC severity was categorized into four groups according to scores of 0, 0–100, 100-399, and > 399 representing as stage normal, mild, moderate, and severe CAD risk. Additionally, the threshold for inclusion in the atherosclerosis group is set at a CAC score of 100 or above” (see Page 4, lines 163-168). We have also included the data on CT angiography in Tables 1 and 2. Notably, following the constructive feedback from the reviewers and the editor, we have revised the term "cardiovascular disease group" to "atherosclerosis group." Patients who had experienced a heart attack were excluded from the study according to the exclusion criteria. However, it is important to note that individuals with cardiovascular disease frequently develop heart attacks.

Changes in the text: Page 4, lines 163-168; Revised Tables 1-2.

  1. The 1hPG and 2hPG might be affected by various factors. Multivariate analyses should be performed.

Reply 4: Thank you for your valuable suggestion. Recognizing that 1hPG and 2hPG may be influenced by various confounding factors, we conducted both univariate and multivariate logistic analyses. These analyses employed unadjusted models, as well as Model 1 (which is adjusted for sex, age, and BMI) and Model 2 (which is further adjusts for triglycerides, HDL-C, uric acid, and ALT) (see Table 3). Additionally, we performed a subgroup analysis based on gender, age, and fasting blood glucose status (see revised Figure 2 and Tables S7-S9). It is important to note that other potential confounding factors, such as smoking, alcohol intake, and caffeine consumption, were not analyzed and are discussed in the limitations (see Page 12, lines 408-410).

Round 2

Reviewer 1 Report

Comments and Suggestions for Authors

Round2

Major

#1, #2, #3, #4. Well-responded.

#5. The authors did not respond to the comment at all. In Table 3, a significant OR on hepatic steatosis was found between the reference and 1hPG(+)&2hPG(-), although there was not a significant difference in OR on hepatic steatosis between the reference and 1hPG(+)&2hPG(+). I guessed that hepatic steatosis was more severe in 1hPG(+)&2hPG(-) than in 1hPG(+)&2hPG(+). However, there was not a significant difference in CT value between the groups of 1hPG(+)&2hPG(-) and 1hPG(+)&2hPG(+). Please explain the reason. In addition, I would like to know the reason why elevated 1hPG level (not 2hPG level) was associated with the severity of hepatic steatosis in MASLD patients. This is one of the most valuable issues obtained from this study.

#6. The authors did not respond to the comment at all. According to Table 3, a significant OR on hepatic fibrosis was present between the reference and 1hPG(+)&2hPG(+), although the OR on hepatic fibrosis was not significant between the reference and 1hPG(-)&2hPG(+). There was a significant difference in FIB4-index between 1hPG(+)&2hPG(-) and 1hPG(+)&2hPG(+) (see Table 2). The authors should discuss the interpretation of these results. In addition, AUC on hepatic fibrosis in 1hPG and that in 2hPG were almost equivalent in Figure 3C. It seems to be strange, because I guessed that AUC in 2hPG was significantly wider than that in 1hPG. Please let me know the reason why 2hPG level (not 1hPG level) was associated with hepatic fibrosis. The result is another important finding in this study.

#7, #8. Well-responded.

Minor

#1, #2, #3, #4, #6, #7, #8. Well-responded

#5. “FIB-4 score” should be corrected to “FIB-4 index” (Page 4, line 162,  Page 7, line 231).

   “ALT (IU/L)^0.5 (Page 4, line 163) should be replaced with “√ALT (IU/L)”.

Comments on the Quality of English Language

The quality of English language may be improved.

Author Response

Dear Editor,

We are most grateful for the time and care the editor and reviewers spent for improving our manuscript.

Please find below the point-by-point responses to your comments. All revisions are highlighted on a yellow background. Additionally, the English has been polished using MDPI Author Services in accordance with the editor's suggestions

Bihui Zhong, MD, PhD

On behalf of the author team

Reviewer #1:

Major

##5. The authors did not respond to the comment at all. In Table 3, a significant OR on hepatic steatosis was found between the reference and 1hPG(+)&2hPG(-), although there was not a significant difference in OR on hepatic steatosis between the reference and 1hPG(+)&2hPG(+). I guessed that hepatic steatosis was more severe in 1hPG(+)&2hPG(-) than in 1hPG(+)&2hPG(+). However, there was not a significant difference in CT value between the groups of 1hPG(+)&2hPG(-) and 1hPG(+)&2hPG(+). Please explain the reason. In addition, I would like to know the reason why elevated 1hPG level (not 2hPG level) was associated with the severity of hepatic steatosis in MASLD patients. This is one of the most valuable issues obtained from this study.

Reply 5: Thank you for your positive feedback and interest in our research. As indicated in Table 3, the 1hPG(+)&2hPG(-) group is identified as a risk factor for moderate-severe liver steatosis in patients with MASLD, compared to the 1hPG(-)&2hPG(-) group. However, Table 2 shows no significant differences in the CT values across the groups. In response to your comment, we would like to clarify that the CT value is a continuous variable that does not follow a normal distribution. Further non-parametric tests have the disadvantages of lower test efficiency and insensitivity to small differences. To address this, as shown in Figure 1B, we categorized the CT values using a cut-off of 40 to differentiate between mild steatosis and moderate-severe hepatic steatosis. A subsequent chi-square test revealed a statistically significant difference in the proportion of moderate-to-severe steatosis across the four groups (P=0.02). Additionally, post hoc analysis using the Bonferroni correction indicated a significant difference between the 1hPG(+)&2hPG(-) and the 1hPG(-) & 2hPG(-) groups (P<0.05).  (see Figure 1B, page 5, lines 211-213). This finding aligns with the results obtained from the logistic regression model.

Furthermore, we have enhanced the Discussion section by incorporating additional details and relevant literature to support our argument regarding why elevated 1hPG levels (but not elevated 2hPG levels) are associated with the severity of hepatic steatosis in patients with MASLD. The revised text states: “Our results suggest that 1hPG levels serve as an earlier indicator of disrupted glucose homeostasis compared to 2hPG levels. This finding is consistent with previous research, which has shown that 1hPG levels are more sensitive to changes in glucose tolerance and insulin secretion than 2hPG levels [43]. Severe fluctuations in blood glucose can lead to insulin resistance, which is characterized by a significant reduction in cellular sensitivity to insulin [44]. This condition results in diminished insulin efficacy in pro-moting glucose uptake and metabolism. Importantly, there is a strong correlation be-tween the degree of insulin resistance and the severity of hepatic fatty degeneration [45]. In this context, 1hPG levels exhibit heightened sensitivity, as they more accurately reflect the extent of hepatic fatty degeneration at an earlier stage, thereby serving as a valuable marker for early assessment and intervention.”  (see Page 12, lines 389-399).

Reference 25: Guerreiro, V, Maia, I, Neves, JS, et al. Oral glucose tolerance testing at 1 h and 2 h: relationship with glucose and cardiometabolic parameters and agreement for pre-diabetes diagnosis in patients with morbid obesity. Diabetol. Metab. Syndr. 2022; 14: 91.

Reference 43: Ferrannini G, De Bacquer D, Gyberg V, et al. Saving time by replacing the standardised two-hour oral glucose tolerance test with a one-hour test: Validation of a new screening algorithm in patients with coronary artery disease from the ESC-EORP Reference 44: Chandrasekaran P, Weiskirchen R. Cellular and Molecular Mechanisms of Insulin Resistance. Curr. Tissue Microenviron. 2024; Rep. 5, 79–90.

Reference 44: Chandrasekaran P, Weiskirchen R. Cellular and Molecular Mechanisms of Insulin Resistance. Curr. Tissue Microenviron. 2024; Rep. 5, 79–90.

Reference 45: Muzurović E, Mikhailidis DP, Mantzoros C. Non-alcoholic fatty liver disease, insulin resistance, metabolic syndrome and their association with vascular risk. Metabolism. Epub 2021; Apr 14.

Changes in the text: Page 5, lines 211-213; Page 12, lines 389-399.

#6. The authors did not respond to the comment at all. According to Table 3, a significant OR on hepatic fibrosis was present between the reference and 1hPG(+)&2hPG(+), although the OR on hepatic fibrosis was not significant between the reference and 1hPG(-)&2hPG(+). There was a significant difference in FIB4-index between 1hPG(+)&2hPG(-) and 1hPG(+)&2hPG(+) (see Table 2). The authors should discuss the interpretation of these results. In addition, AUC on hepatic fibrosis in 1hPG and that in 2hPG were almost equivalent in Figure 3C. It seems to be strange, because I guessed that AUC in 2hPG was significantly wider than that in 1hPG. Please let me know the reason why 2hPG level (not 1hPG level) was associated with hepatic fibrosis. The result is another important finding in this study.

Reply 6: Thank you for your valuable advice. No significant difference in the FIB-4 index was observed between the 1hPG(-)&2hPG(+) and 1hPG(-)&2hPG(-) groups. However, the FIB-4 index was categorized into two distinct groups based on a threshold of 1.3, and the Chi-square test revealed a statistically significant difference in the proportion of significant fibrosis among the four groups, with a p-value of 0.01. The post hoc analysis, utilizing the Bonferroni correction, indicated that the proportion of significant fibrosis was higher in the 1hPG(+)&2hPG(+) group compared to the 1hPG(-)&2hPG(-) group (see Figure 1B), which was consistent with the crude model presented in Table 3. Notably, after adjusting for potential confounders including sex, BMI, triglycerides, HDL-C, and uric acid, the 1hPG(-)&2hPG(+) group emerged as a significant risk factor for liver fibrosis when the 1hPG(-)&2hPG(-) group was used as a reference. This suggests that the association between 2hPG elevation and liver fibrosis is influenced by these confounding factors. Similarly, the comparable AUC for hepatic fibrosis in 1hPG and 2hPG can be attributed to our failure to adjust for potential confounders, including sex, BMI, triglycerides, HDL-C, and uric acid.

What’s more, we have further enhanced the content in the discussion section and included relevant references to support our arguments. The revised text states: “Our results indicate a significant association between 2hPG and liver fibrosis. One possible explanation for this finding is that the sensitivity of 2hPG is lower than that of 1hPG in detecting changes in disrupted glucose homeostasis [43]. Consequently, an abnormal 2hPG level may reflect a more severe degree of liver injury, potentially indicating the presence of significant liver fibrosis or other pathological changes [48].”(see page 12, lines 408-413). Thanks again for your advice.

Reference 43: Ferrannini G, De Bacquer D, Gyberg V, et al. Saving time by replacing the standardised two-hour oral glucose tolerance test with a one-hour test: Validation of a new screening algorithm in patients with coronary artery disease from the ESC-EORP Reference 44: Chandrasekaran P, Weiskirchen R. Cellular and Molecular Mechanisms of Insulin Resistance. Curr. Tissue Microenviron. 2024; Rep. 5, 79–90.

Reference 48: Ferrannini G, De Bacquer D, Gyberg V, et al. Saving time by replacing the standardised two-hour oral glucose tolerance test with a one-hour test: Validation of a new screening algorithm in patients with coronary artery disease from the ESC-EORP EUROASPIRE V registry. Diabetes Res Clin Pract. 2022; Jan;183:109156.

Changes in the text: Page 12, lines 408-413.

Minor
#5. “FIB-4 score” should be corrected to “FIB-4 index” (Page 4, line 162,  Page 7, line 231).

   “ALT (IU/L)^0.5 (Page 4, line 163) should be replaced with “√ALT (IU/L)”.

Reply 5: Thank you for your kind reminder. We have revised the term “FIB-4 score” to “FIB-4 index” throughout the text. Additionally, we are sorry for error; ALT (IU/L)^0.5 has been replaced with “√ALT (IU/L)”.

Changes in the text: Page 4, lines 161-162. Page 7, line 231.

Reviewer 3 Report

Comments and Suggestions for Authors

Revised manuscript was well-addressed to the reviewer's comments and well-written.

Author Response

Dear Editor,

We are most grateful for the time and care the editor and reviewers spent for improving our manuscript.

Bihui Zhong, MD, PhD

On behalf of the author team